# c-Met^+^ Cytotoxic T Lymphocytes Exhibit Enhanced Cytotoxicity in Mice and Humans In Vitro Tumor Models

**DOI:** 10.3390/biomedicines11123123

**Published:** 2023-11-23

**Authors:** Mahdia Benkhoucha, Ngoc Lan Tran, Isis Senoner, Gautier Breville, Hajer Fritah, Denis Migliorini, Valérie Dutoit, Patrice H. Lalive

**Affiliations:** 1Department of Pathology and Immunology, Faculty of Medicine, University of Geneva, 1211 Geneva, Switzerland; mahdia.benkhoucha@unige.ch (M.B.); lan.tran@unige.ch (N.L.T.); isis.senoner@unige.ch (I.S.); hajer.fritah@unige.ch (H.F.); 2Department of Clinical Neurosciences, Division of Neurology, University Hospital of Geneva, 1205 Geneva, Switzerland; gautier.breville@pennmedicine.upenn.edu; 3Center for Neuroinflammation and Experimental Therapeutics, Department of Neurology, Perelman School of Medicine, University of Pennsylvania, Philadelphia, PA 19104, USA; 4Brain Tumor and Immune Cell Engineering Laboratory, Department of Medicine, Faculty of Medicine, University of Geneva, 1211 Geneva, Switzerland; denis.migliorini@hcuge.ch (D.M.); valerie.dutoit@hcuge.ch (V.D.); 5Department of Oncology, Unit of Neuro-Oncology, University Hospital of Geneva, 1205 Geneva, Switzerland

**Keywords:** c-Met, HGF, CTLs, CD8^+^ cytotoxic T lymphocytes, spheroids, tumor

## Abstract

CD8^+^ cytotoxic T lymphocytes (CTLs) play a crucial role in anti-tumor immunity. In a previous study, we identified a subset of murine effector CTLs expressing the hepatocyte growth factor (HGF) receptor, c-Met (c-Met^+^ CTLs), that are endowed with enhanced cytolytic capacity. HGF directly inhibited the cytolytic function of c-Met^+^ CTLs, both in 2D in vitro assays and in vivo, leading to reduced T cell responses against metastatic melanoma. To further investigate the role of c-Met^+^ CTLs in a three-dimensional (3D) setting, we studied their function within B16 melanoma spheroids and examined the impact of cell–cell contact on the modulation of inhibitory checkpoint molecules’ expression, such as KLRG1, PD-1, and CTLA-4. Additionally, we evaluated the cytolytic capacity of human CTL clones expressing c-Met (c-Met^+^) and compared it to c-Met^−^ CTL clones. Our results indicated that, similar to their murine counterparts, c-Met^+^ human CTL clones exhibited increased cytolytic activity compared to c-Met^−^ CTL clones, and this enhanced function was negatively regulated by the presence of HGF. Taken together, our findings highlight the potential of targeting the HGF/c-Met pathway to modulate CTL-mediated anti-tumor immunity. This research holds promise for developing strategies to enhance the effectiveness of CTL-based immunotherapies against cancer.

## 1. Introduction

Malignant tumors exist in a complex microenvironment known as the tumor microenvironment (TME), composed of various cell types, including stromal cells, neutrophils, dendritic cells (DCs), and macrophages. Some of these TME-resident cells release hepatocyte growth factor (HGF) and promote the activation of c-Met, the receptor for HGF, within tumor cells [1,2]. c-Met expression has been linked to poor prognosis in different solid tumors [3,4,5], and the overactivation of the HGF/c-Met axis has been shown to promote tumorigenesis and tumor progression in various cancer types, such as gastroesophageal adenocarcinoma, cholangiocarcinoma, colon cancer, kidney cancer, glioblastoma, melanoma, and lung cancer [6,7,8,9,10,11,12,13,14]. Some studies suggest that HGF can act as an immunotolerant factor in autoimmune diseases by affecting DCs [15,16,17,18]. By contrast, others studies report that HGF can be a potent negative regulator of DC function by (i) stimulating T regulatory lymphocytes (Tregs), (ii) decreasing interleukin-17-producing lymphocytes [15], and (iii) increasing the production of both interleukin-10 and transforming growth factor beta [16]. In addition, c-Met deletion in neutrophils enhances tumor growth and metastasis, as c-Met is required for chemoattraction and neutrophil-mediated cytotoxicity [19]. In blood samples, this phenomenon is represented by a correlation between c-Met deletion and reduced neutrophil infiltration to both primary tumors and distant metastases [19]. Finally, HGF acts on myeloid-derived suppressor cells by inducing their expansion and increasing iNOS and ARG1 expression, as well as harboring a suppressive function and inducing Tregs [20]. Limited studies have reported the expression of c-Met on murine CD4^+^ and CD8^+^ thymocytes [21] and T cells [22].

In the context of T lymphocytes, we identified a novel population of highly cytotoxic CTLs expressing c-Met (c-Met^+^ CTLs) in a murine model of lung metastasis [23]. These c-Met^+^ CTLs exhibited enhanced cytolytic activity with an increased production of interferon (IFN-)γ and granzyme (Gr) B compared to their c-Met^−^ counterparts [23]. Furthermore, we demonstrated the immunosuppressive role of HGF in c-Met^+^ CTLs both in vitro and in vivo [23]. Adoptive transfer of highly cytotoxic CD8^+^c-Met^+^ T cells in tumor-bearing mice induced significantly tumor growth suppression as compared to CD8^+^c-Met^−^ T cells. Additionally, we identified CD8^+^c-Met^+^ tumor-infiltrating lymphocytes (TILs) in human melanoma skin biopsies [23], suggesting that targeting the HGF/c-Met pathway could control CD8^+^c-Met^+^ T cell-mediated anti-tumor immunity.

Immune checkpoint inhibitors (ICIs) have shown promising results in cancer treatment; in particular, ICIs targeting CTLA-4, PD-1, and PD-L1 have yielded unprecedented and durable responses in a significant percentage of cancer patients in recent years, leading to the US Food and Drug Administration (FDA) approval of six ICIs for numerous cancer indications since 2011 [24]. However, they are efficient only in a subset of patients and are associated with severe side effects [25]. As an alternative approach, cell therapy, such as TILs, engineered T cell receptor (TCR), chimeric antigen receptor (CAR)-T cells, CTLs, and natural killer (NK) cell therapy, have emerged as advanced strategies to overcome the limitations faced by ICIs [24,26]. Nevertheless, effective cellular therapies in solid tumors encounter challenges related to target antigen identification, T cell trafficking to the tumor site, and the immunosuppressive TME. Combining cellular therapies with ICIs may improve T cell infiltration and enhance anti-tumor responses [27,28].

Based on our recent findings [23], we hypothesized that the oncogenic protein c-Met may directly regulate CTL effector functions in the B16 melanoma spheroid model and human CTL clones. In this study, we focused on the phenotypic and functional modulation of c-Met^+^ CTLs generated from Pmel-1 mice in the 3D tumor culture, under stress cell–cell and cell–matrix interaction, which cannot be studied in 2D models. Indeed, cells growing in spheroids display a higher degree of functional and morphological differentiation than monolayer cells. As a result, spheroids can mimic the in vivo spatial architecture, physiological responses, gene expression patterns, mechanisms of drug resistance, and secretion of molecules induced through cell stimuli [29]. We thus employed a 3D culture model to partially mimic aspects of the TME and co-cultured B16 melanoma cell-derived spheroids with CTLs c-Met^+^ to investigate the impact of tumor cells on immune cell effector functions, and vice versa [30]. We also explored, for the first time, the role of c-Met expression in human CTLs. We used human CTL clones to study c-Met expression and its impact on cytolytic function, along with the effect of HGF modulation on their activity. We found that 3D B16 spheroids influenced the expression of negative costimulatory molecules, such as KLRG1, PD-1, and CTLA-4, on c-Met^+^ CTLs. Finally, we showed, for the first time, that c-Met expression in human CTL clones conferred higher cytolytic capacity, while HGF treatment dampened this activity by reducing GrB and IFN-γ production.

## 2. Methods

### 2.1. Reagents

Human recombinant HGF (hHGF) was used throughout (#100-39H-25UG; peprotech), as human and murine HGF are cross-reactive. The dose of hHGF (30 ng/mL) used was chosen based on previous studies analyzing the immunoregulatory effects of hHGF on CD4^+^ T cell responses [15]. Lower doses were not as effective.

### 2.2. Spheroids Preparation

Generations of 3D spheroids from B16 melanoma cells can be co-cultured with immune cell populations of interest; we directly cultured B16 cells in wells precoated with agarose. The melanoma B16 cells were harvested and counted, and 1000 cells were resuspended in 200 μL of DMEM supplemented with 10% FCS, 40 μg/mL of streptomycin, 40 units/mL of penicillin, and 200 mM of l-glutamine, and were plated in each well of the 96-well flat-bottom plates (Corning, Corning, NY, USA) precoated with 1% agarose (Thermo Fisher, Waltham, MA, USA) dissolved in double-distilled water. Then, the spheroids were incubated for 7 days in a 5% CO_2_ humidified incubator at 37 °C. All experiments were performed on that day, and the day was assigned as day 0. The control spheroids were digested with trypsine-0.05% EDTA (#25300054, Thermo Fisher, Waltham, MA, USA) and analyzed with flow cytometry to visualize the presence of necrosis in the untreated samples (absence of CTLs).

### 2.3. Cells and Cultures

Splenocytes were derived from Pmel-1 T cell receptor (TCR) transgenic mice (The Jackson Laboratory, Bar Harbor, ME, USA), whose TCR recognizes an H-2D^b^-restricted epitope corresponding to amino acids 25–33 of murine and human gp100 (gp100_25–33_), a self-tumor antigen. Approximately 90% of the splenic CD3^+^ T cells in Pmel-1 TCR mice are TCRVβ13^+^ CD8^+^ T cells and demonstrate specificity for gp100 [31].

### 2.4. CTL Stimulation

A suspension of individual cells was created from the combined spleens of the Pmel-1 mice. This was achieved by gently breaking down the tissues and filtering them through a 70 μm Nylon cell strainer (Falcon, BD Biosciences, San Jose, CA, USA). To remove red blood cells, an ACK lysing buffer (BioWhittaker, Rockland, ME, USA) was employed. The cells were then suspended in RPMI and activated with 10 μg/mL of Gp100_25–33_ peptide (KVPRNQDWL) for 1 h at 37 °C. After washing and counting, IL-2 (50 U/mL; Pepro-Tech, Cranbury, NJ, USA) was introduced into the cultures on days 2 and 4. Cells that were cultured for 5 days underwent functional and phenotypic testing.

### 2.5. Human CTL Clones and T2 Cells

Human CTL clones and T2 cells were maintained in RPMI 1640 supplemented with 200 mM of l-glutamine, 50 μM of β-mercaptoethanol, 100 mM of sodium pyruvate, MEM vitamins, 40 μg/mL of streptomycin, 40 units/mL of penicillin (standard medium M′), and 8% human FCS (Laboratoires Jacques Boy, Reims, France). Human CTL clones targeting different antigens were kindly proposed by the laboratory of Pierre Yves Dietrich (Geneva University Hospitals, Geneva, Switzerland). Briefly, the human clones where isolated using peptide-MHC tetramers and cultured in limiting dilution condition as previously described [32]. T cell clones were periodically restimulated, as previously described [32]. We used the following clones: clones GE 450 1E8 and GE 522 5E3, specific for the HLA-A2-restricted Melan-A (*MLA*)-derived peptide (26–35 A27L), from patients with melanoma, and clone GE 549 1F10, specific for the HLA-A2-restricted Brevican (*BCA2*)-derived peptide (478–486), from patients with glioblastoma [32]. T2 cells are HLA-A*0201 human lymphoid cells that are defective in antigen processing but effectively present exogenously supplied peptides [33].

### 2.6. Flow Cytometry

Single-cell suspensions from Pmel-1 spleens generated in vitro were incubated in a blocking solution (PBS with 1% fetal calf serum) for 20 min on ice prior to staining to block non-specific Fc-mediated interactions. Then, they were stained for 30 min at 4 °C with appropriate fluorochrome-conjugated antibodies (Abs, 1:100) (Appendix A) or isotype-matched irrelevant Abs to determine background fluorescence. For intracellular cytokines and the molecular staining of IFN-γ, TNF-α, and GrB, the T cells were stimulated with phorbol myristate acetate (PMA, 50 ng/mL) and ionomycin (500 ng/mL) in the presence of brefeldin A (10 μg/mL; Sigma-Aldrich, Burlington, MA, USA) and then fixed and permeabilized using a BD Cytofix/Cytoperm Plus Kit (BD Biosciences). The samples were processed on a FACS LSRFortessa flow cytometer (BD Biosciences) and analyzed using the FlowJo analysis software (version 10.3). Live, apoptotic, and dead populations were defined based on a 7-AAD Viability Staining Solution from eBioscience or with 0.2 ug/mL of Dapi (#D1306) and Annexin V (#17-8007-72), according to the manufacturer’s instructions.

Cytotoxicity was determined as the level of necrotic (Dapi^+^AnnexinV^−^) cells relative to the negative control condition (necrotic B16 cultured alone in the absence of CTLs). The percent of specific lysis was calculated according to the formula (1 − (control/test)) × 100. To determine the viability of the cells prior to fixation and permeabilization, we used the LIVE/DEAD™ Fixable Aqua Dead Cell Stain Kit (Thermo Fisher Scientific).

For cell separation, at the end of culture, the cells were harvested, stained at 4 °C for 30 min with CD3, CD8, and c-Met, and purified as CD8^+^ c-Met^−^ and CD8^+^ c-Met^+^ through cell sorting using a FACS Aria SORP II cell sorter (BD Biosciences). Live cells were gated based on forward scatter (FSC) and side scatter (SSC) and through DAPI exclusion. The Ag-activated c-Met^−^ and c-Met^+^CD8^+^ T cell populations showed similar physical properties of size and internal complexity. The cell viability (routinely >95%) at the end of the FACS sorting procedure was determined using the trypan blue dye exclusion method.

A human Fc block (BD Pharmingen, San Diego, CA, USA) was used to block the non-specific binding of Fc receptors prior to the extracellular staining of human CTL clones. They were then stained for 30 min at 4 °C with appropriate fluorochrome-conjugated Abs (1:100) (Appendix A) or isotype-matched irrelevant Abs to determine background fluorescence. For intracellular cytokines and the molecular staining of IFN-γ and GrB, the T cells were stimulated and processed as explained before.

### 2.7. RT-PCR

The human CTL clones were treated with hHGF (30 ng/mL) for 72 h. Total RNA was extracted using the RNeasy micro kit from Qiagen, followed by quantitative real-time duplex PCR analysis using the ABI 7500 system by Applied Biosystems. Prior to this analysis, reverse transcription was carried out using the iScript cDNA Synthesis Kit from Bio-Rad. The primers and probes used in this analysis were sourced from Applied Biosystems. To standardize the mRNA expressions of *MET*, *IFNG*, *GZMB*, *PRF1*, *TNF*, *FASL*, and *PD-1*, they were normalized with the concurrent analysis of a housekeeping gene *ACTB*. All measurements were performed in triplicate.

### 2.8. In Vitro Cytotoxicity Assay

The cytotoxicity of the antigen-specific human CTL clones was analyzed with a DELFIA^®^ cell cytotoxicity kit (PerkinElmer, Waltham, MA, USA) according to the manufacturer’s instructions. Briefly, the CTL clones were either incubated or not with HGF for 72 h (30 ng/mL) before use in the assay. T2 target cells (10^5^ cells) were either incubated or not with the cognate CEF3, CEF5, MLA, or BCA2 peptides for 1 h at 37 °C; the rest is identical to the method that we have published elsewhere [23]. The percent of specific lysis was calculated according to the formula (experimental release − spontaneous release)/(maximum release − spontaneous release) × 100.

### 2.9. Western Blotting

Human BCA2-specific CTLs were homogenized using a polytron in a lysis buffer (50 mM of Tris–HCl (pH 7.5), 250 mM of NaCl, 1% Triton X-100, 1 mM of EDTA, and 1 mM of DTT) containing complete protease inhibitors (Roche, Basel, Switzerland). Equal amounts (20 μg) of the total protein from each sample were transferred to 8% sodium dodecyl sulfate (SDS)–polyacrylamide gel and blotted onto an Immobilon-P polyvinylidene difluoride (PVDF) membrane (Millipore, Burlington, MA, USA). Antibodies for Phospho-c-Met (Tyr1234/5, #3077, 1:500), c-Met (#3127, 1:400), and β-actin (#8457, 1:1000) were all obtained from Cell Signaling Technology (Danvers, MA, USA). The membranes were cut and the two parts were incubated overnight at 4 °C with either the c-Met (~145–170 kDa) or the β-actin (~42 kDa) primary antibody. The same procedure was followed to detect the phospho c-Met (~140 kDa). Next, the membranes were washed with TBS/T and incubated with anti-mouse IgG (Cell Signaling #7076) or anti-rabbit IgG (Cell Signaling #7074) horseradish peroxidase-conjugated secondary antibody for 1 h at room temperature. Bands were developed by adding horseradish peroxidase substrate (Thermo Fisher) and recorded with X-ray film or the ChemiDoc^TM^ digital imaging system from Bio-Rad.

### 2.10. Statistical Analysis

Statistical comparisons were performed using paired and unpaired Student’s *t*-tests. *p* < 0.05 was considered statistically significant. All statistical analyses were performed with the GraphPad Prism software, version 9.2.

## 3. Results

### 3.1. Using 3D B16 Spheroids to Study the Phenotype and Function of Tumor-Specific c-Met^+^ CTLs

Here, we provide a detailed workflow used for the generation of 3D spheroids from B16 melanoma cells that can be co-cultured with cytotoxic cells from Pmel-1 mice. By using agarose pre-coated wells and seeding B16 cells on the top, spheroids were generated over a period of 7 days, as shown in Figure 1A. After in vitro activation, the CTLs were sorted into c-Met^+^ and c-Met^−^ cells (Appendix A) and co-cultured with B16 spheroids for 4 h to determine their cytotoxic capacity and phenotype modification in contact with the spheroids. We named adherent CTLs (c-Met^+^ or c-Met^−^), CTLs that were in contact with the spheroids even after washing with PBS, and non-adherent CTLs (Figure 1B).

### 3.2. Phenotypic and Function of c-Met^+^ CTLs in Contact with 3D B16 Spheroids

Our previous data indicated that cytotoxic T cells expressing c-Met (c-Met^+^CTLs) are highly cytotoxic compared to their c-Met^−^ counterpart [23]. We anticipated a similar cytolytic performance on the 3D spheroid culture. In particular, the 3D spheroids were shown to be markedly more susceptible to cell death than cells grown in a monolayer (2D) conformation [34]. We tested the lysis function via flow cytometry and by quantifying the necrotic B16 DAPI^+^, as shown in the gating strategy (Figure 1B, right panel). We found that the rate of B16 spheroid lysis was significantly increased when co-cultured with c-Met^+^ CTLs compared to with their c-Met^−^ counterpart (Figure 2A). We then assessed the proportion of CTLs producing IFN-γ, GrB, and TNF-α in contact with the spheroids, and showed that the c-Met^+^ T cells expressed higher levels of these three molecules compared to their c-Met^−^ counterpart (Figure 2B and Appendix A for isotype controls). Interestingly, it was also observed in the non-adherent CTL compartment. Notably, in contact with the B16 spheroids, the c-Met^+^ CTLs displayed increased CD107a expression, a marker commonly used to measure CTL activity (Figure 2C). The analysis of FasL (CD178) expression did not show any difference between both populations in contact with B16 spheroids (Appendix A).

### 3.3. Modulation of Immune Checkpoint Expression by c-Met^+^ CTLs in Contact with 3D B16 Spheroids

We next assessed whether cell contact between CTLs and B16 3D spheroids modulated immune checkpoint molecules such as KLRG1, PD-1, CTLA-4, Tim-3, and LAG-3. We observed a significant increase in KLRG1 expression on the c-Met^+^ CTLs compared to their c-Met^−^ counterpart (Figure 3A and Appendix A for isotype controls). This observation is in line with another publication showing that, during acute infections or vaccinations, naïve CD8^+^ T cells become activated and differentiate into effector T cells containing KLRG1^Hi^ terminal effector cells [35]. In contrast, we detected a decreased expression of inhibitory receptors such as PD-1 and CTLA-4 on the c-Met^+^ CTLs compared to their c-Met^−^ counterpart upon contact with the B16 spheroids (Figure 3B,C and Appendix A for isotype controls). Additionally, other checkpoint molecules such as Tim-3 and LAG-3 were not modulated in either population upon contact with the B16 spheroids (Appendix A).

### 3.4. Modulation of c-Met Expression, Phenotype, and Function by HGF on Human T Cell Clones

Then, we assessed the expression of c-Met on tumor antigen-specific human CD8^+^ T cell clones generated from cancer patients. We observed a variable expression of c-Met in the different clones, both at the mRNA and protein levels (Figure 4A,B). Interestingly, the clones that displayed significant c-Met expression (GE 522 5E3 and GE 549 1F10) had an efficient killing capacity of T2 target cells loaded with the specific antigen (*MLA* or *BCA2*), which was reduced after the addition of human recombinant (h)HGF (Figure 4C). To further study the biological function of hHGF on c-Met-expressing CD8 clones, we selected the GE 549 1F10 clone which showed the highest reduction in killing activity in the presence of hHGF (Figure 4C), and we confirmed the presence of phosphor-c-Met (P-c-Met) one hour after hHGF treatment (Figure 4D). Additionally, using RT-PCR on the clone of interest, we assessed its phenotypic profile in the presence of hHGF. We noticed a significant decrease in the IFN-γ and TNF-α expression levels and, although less striking, a similar tendency was observed for the GrB, perforin, FasL, and PD-1 levels (Figure 5A). At the protein level, the production of IFN-γ and GrB was reduced after incubation with hHGF (Figure 5B,C). Altogether, our data suggest the presence of cytotoxic c-Met^+^ CD8 clones derived from cancer patients, and these cells have a higher cytotoxic capacity compared to c-Met^−^ CD8 clones, which is decreased upon in vitro treatment with hHGF.

## 4. Discussion

The receptor tyrosine kinase c-Met plays a pivotal role in embryonic development and tissue regeneration, and influences critical cellular processes such as cell survival, proliferation, migration, and angiogenesis [36] The HGF/c-Met axis actively participates in the regulation of numerous inflammatory-mediated diseases by engaging a diverse array of cell types [37]. Extensive research has established the HGF/c-Met signaling pathway as a promising target for personalized cancer treatment, especially when expressed on tumor cells [38,39]. However, it is worth noting that only patients with tumors overexpressing c-Met can benefit from HGF/c-Met targeting therapy, as these interventions inhibit typical c-Met-associated processes such as oncogenesis, cancer metastasis, and drug resistance [40]. Our previous research demonstrated that HGF treatment reduced the effector capacity of tumor-specific CD8^+^ T cells in a murine model of CTL-mediated killing [33]. These findings highlighted the pivotal role of c-Met expression on tumor-infiltrating CD8^+^ T cells in enhancing their anti-tumoral functions and restraining tumor growth [23]. Nonetheless, the existence of c-Met-expressing CTLs in human settings, as well as the mechanisms governing their functions in response to HGF within the TME, remains unanswered.

This study aims to shed light on these questions using an in vitro model involving B16 melanoma spheroids, which offers a relevant and adaptable system for exploring the interplay between c-Met^+^ CTLs and tumor cells. Our observations revealed the presence of a tumorigenic CD8^+^ T cell subtype expressing c-Met with increased activation levels when in direct contact with B16 spheroids (adherent CTLs) compared to non-adherent c-Met^+^ CD8^+^ T cells and c-Met^−^ CD8^+^ T cells, consistent with our previous findings regarding murine c-Met^+^ CTLs [23,41]. Furthermore, c-Met^+^ CTLs in contact with B16 spheroids exhibited reduced levels of inhibitory receptors (e.g., PD-1, CTLA-4) compared to their c-Met^−^ CTL counterparts. Together, these data indicate that the expression of c-Met by cytotoxic T lymphocytes (CTLs) may confer a protective advantage against the immunosuppressive influence of the tumor microenvironment. One plausible hypothesis to explain this phenomenon is that the direct interaction between c-Met-expressing CD8^+^ T cells and tumor cells fosters a feedback loop, resulting in an upregulation of c-Met expression on the surface of CTLs together with a reduction in inhibitory molecules, subsequently enhancing their activation. Nevertheless, the precise mechanisms underpinning this intricate process remain to be fully elucidated.

In our previous findings, we successfully demonstrated the presence of c-Met-expressing cytotoxic T lymphocytes in an in vitro murine model of melanoma. Building upon these results, we proceeded to explore the presence of these cells in human PBMCs in an in vitro setting. We demonstrated, for the first time, that human CD8^+^ T cell clones derived in vitro from cancer patients’ PBMCs expressed the c-Met receptor to varying degrees upon TCR activation. Most importantly, c-Met^+^ CD8^+^ clones produced higher levels of effector molecules (e.g., IFN-γ and GrB) compared to their c-Met^−^ CD8^+^ counterparts and exhibited significant specific lysis capacities against tumor cells. These higher levels decreased when treated with HGF, suggesting a direct influence of HGF on CTL-mediated cytotoxicity based on dual GrB/IFN-γ. Altogether, our data suggest the presence of cytotoxic c-Met^+^ CD8^+^ clones derived from cancer patients, and these cells have a higher cytotoxic capacity compared to c-Met^−^ CD8^+^ clones, which is decreased upon in vitro treatment with HGF.

Thus, the collective findings underscore the potential therapeutic implications of targeting the HGF/c-Met signaling axis, particularly in tumors that overexpress c-Met. Currently, three primary methods are employed to inhibit the kinase activity of c-Met:Preventing the extracellular binding of HGF through neutralizing antibodies or biological antagonists;Blocking the phosphorylation of tyrosine residues in the kinase domain with small-molecule inhibitors;Disrupting c-Met kinase-dependent signaling through relevant signal transducers or downstream signaling components [42].

Various small-molecule inhibitors and monoclonal antibodies that recognize c-Met have undergone evaluation in preclinical studies, such as capmatinib, a highly potent and selective c-Met inhibitor recently approved by the FDA for treating metastatic non-small-cell lung cancer [34]. However, in light of our study, it is crucial to consider the potential repercussions of these therapies on cytotoxic T lymphocytes that express c-Met within the tumor microenvironment.

One potential limitation of this study is the use of in vitro B16 organoids rather than an in vivo model, to replicate the complexity of the tumor microenvironment (TME). However, our choice was driven by our team’s prior work with an in vivo model of B16 lung metastasis [23], revealing a substantial presence of c-Met^+^ CTLs in the TME with a robust cytotoxic profile. One of our aims was to demonstrate the feasibility of working with these cells in an in vitro context, bridging the gap to human applications and aligning with the principles of the 3R rule, which promotes the reduction and replacement of animal use with alternative methods. In this case, we believe that the use of B16 3D spheroids is a suitable alternative.

Another potential limitation pertains to the limited number of human clones featured in this study, primarily due to a restricted access to clinical material. Additional mouse studies and expanded human cohorts will be essential to further validate our current findings.

In conclusion, our study presents compelling evidence of the potent cytotoxicity of c-Met^+^ CTLs in both murine and human in vitro models, while also emphasizing the potential influence of HGF on the functionality of these CTLs. These findings underscore the importance of further research into the role of c-Met^+^ TILs in anti-tumor immunity and the implications of HGF/c-Met targeting therapies in the context of CTL-mediated anti-tumoral responses. A deeper understanding of these mechanisms will potentially pave the way for the development of more effective and targeted immunotherapies for cancer treatment.

## Figures and Tables

**Figure 1 biomedicines-11-03123-f001:**
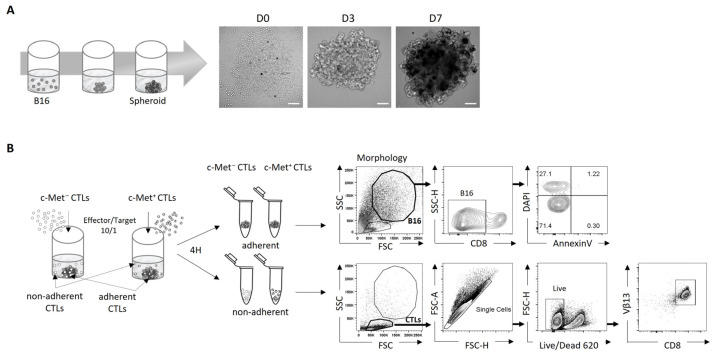
Schematic diagram of protocol for B16 spheroid formation and cytotoxic in vitro test. (**A**) Representative images on the right show the aggregates formed by B16 melanoma cells (1000 cells seeded) over 7 days. The images were acquired with a Zeiss Observer Z1 microscope. Scale bar, 200 µm. (**B**) Diagram illustrating the different steps of 3D B16 culture process model with c-Met^+^ and c-Met^−^ CTLs (left panel), and the gating strategy for B16 cells and CTLs (right panel).

**Figure 2 biomedicines-11-03123-f002:**
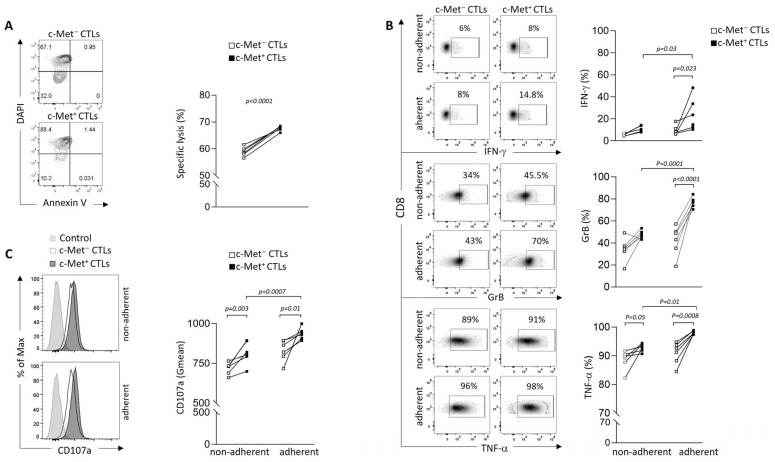
Phenotypic and functional characterization of murine melanoma c-Met^+^ and c-Met^−^ CTLs in 3D B16 spheroid culture. (**A**) In vitro cytolytic activity of day 5-activated c-Met^+^ and c-Met^−^ Pmel-1 CD3^+^CD8^+^ T cells. Effector/target ratios of 10/1 were tested for the killing assay, B16 alone was used as control. Annexin V and DAPI were used to determine the percentage of necrotic B16 cells as shown with contour plot (left panel) and data are represented as paired samples between c-Met^−^ vs. c-Met^+^ CTLs for each mouse (right panel). Cytotoxicity was determined as level of necrotic (Dapi^+^AnnexinV^−^) cells relative to negative control condition (necrotic B16 in absence of CTLs). Values represent mean ± SEM of six mice (*n* = 6) per group. *p*-values were calculated using one-way ANOVA with Tukey’s multiple comparisons test. (**B**) Representative density plots (left panel) and paired flow cytometry quantifications (right panel) of IFN-γ, GrB, and TNFα in CD8^+^c-Met^+^ vs. CD8^+^c-Met^−^ T cells from CTLs adherent to spheroids or CTLs in suspension (non-adherent) (*n* = 6). *p*-values were calculated using a paired *t*-test, and unpaired *t*-test for intergroup comparison. (**C**) Representative histogram plots and average geometric mean (right panel) of CD107a 4 h after co-culture of spheroids B16 and CTLs c-Met^+^ vs. c-Met^−^ from six mice (*n* = 6). *p*-values were calculated with one-way ANOVA with Tukey’s multiple comparisons test. Data of figures (**A**–**C**) are representative of three independent experiments.

**Figure 3 biomedicines-11-03123-f003:**
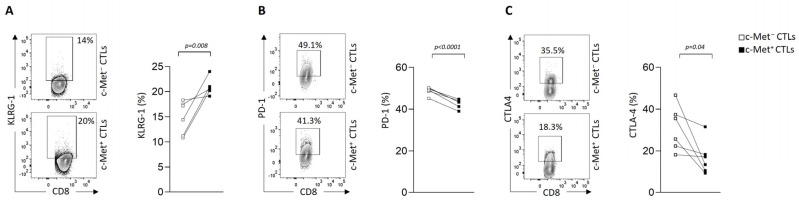
Immune checkpoint molecules’ modulation of c-Met^+^ vs. c-Met^−^ CTLs in contact with 3D B16 spheroids. (**A**–**C**) Representative contour plots (left panel) and paired flow cytometry quantifications (right panels) of KLRG1, PD1, and CTLA-4 expression on c-Met^+^ vs. c-Met^−^ CTLs, 4 h after co-culture with B16 spheroids (*n* = 6 mice). *p*-values were calculated using a paired *t*-test. Data of figures (**A**–**C**) are representative of three independent experiments.

**Figure 4 biomedicines-11-03123-f004:**
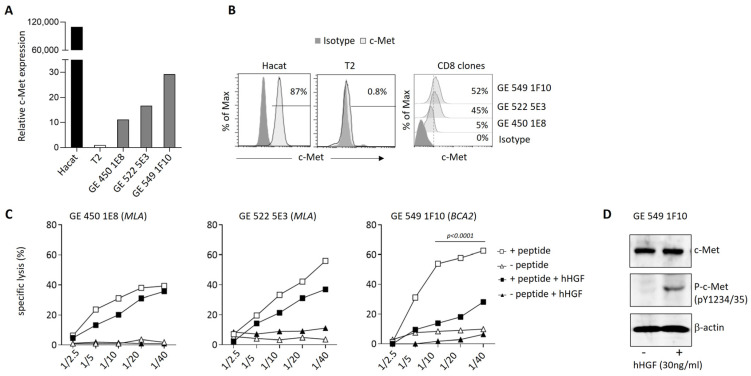
c-Met expression on human CTL clones and modulation of killing capacity by HGF in vitro. (**A**) c-Met mRNA expression via qRT-PCR from CTL clones isolated from cancer patients compared to Hacat and T2 cell lines, used as positive and negative controls, respectively. (**B**) Flow cytometry histograms showing c-Met protein expression (light gray) on CTL clones and control cell lines compared to the isotype control (dark gray). Percentages of c-Met^+^ CTLs are indicated. (**C**) In vitro killing assays were performed using DELFIA^®^ cell cytotoxicity kit (PerkinElmer). T2 target cells were cultured with indicated ratio of CTL clones in the presence (squares) or absence (triangles) of the cognate peptide. Clones were incubated or not with HGF for 72 h (black symbols, 30 ng/mL) before use in the assay. MLA: HLA-A2-restricted Melan-A-derived peptide (26–35 A27L), BCA2: HLA-A2-restricted Brevican-derived peptide (478–486). Each point corresponds to the value of the average of the triplicates; this experiment is representative of three independent experiments. (**D**) Protein expression levels of c-Met and phosphor c-Met from BCA2 clone 1 h after treatment or not with hHGF, as measured with Western blot. Data of figures A, B, and C are representative of three independent experiments of *n* = 3. *p*-values were calculated using one-way ANOVA with Tukey’s multiple comparisons test.

**Figure 5 biomedicines-11-03123-f005:**
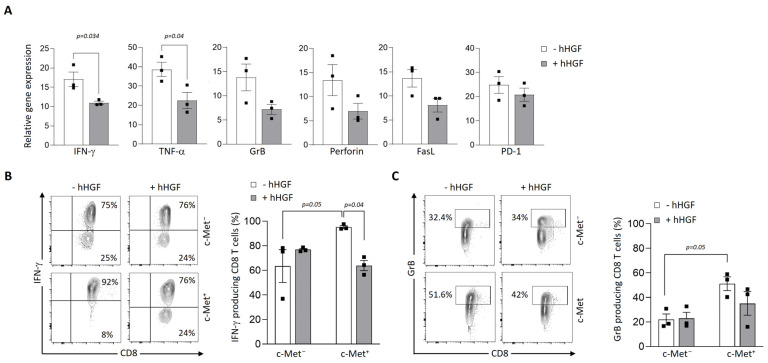
HGF modulation of GBM-specific clone function. (**A**) Relative expression of *IFNG*, *TNF*, *GZMB*, *FASL*, and *PD-1* mRNA in the BCA2-specific clone GE 549 1F10 72 h after treatment or not with hHGF, as measured via qRT–PCR. Data are representative of three independent experiments of *n* = 3 carried out in triplicate. Error bars show mean ± SEM, *p*-values were calculated using unpaired 2-tailed Student’s *t*-test. (**B**,**C**) Representative contour plots of IFNγ and GrB (left panels) and mean frequencies (bar graphs) of the BCA2-specific clone GE 549 1F10 expressing c-Met or not after treatment or not with hHGF for 72 h (right panel). *p*-values were calculated using one-way ANOVA with Tukey’s multiple comparisons test. Data are representative of three independent experiments of *n* = 3.

## Data Availability

Data are contained within the article.

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
