# Peer review of "c-Met^+^ Cytotoxic T Lymphocytes Exhibit Enhanced Cytotoxicity in Mice and Humans In Vitro Tumor Models"

_biomedicines, 2023, doi:10.3390/biomedicines11123123_

Round 1
Reviewer 1 Report
Comments and Suggestions for Authors
In the manuscript entitled “c-MET+ CTLs exhibit enhanced cytotoxicity in mouse and human in vitro tumor models”, the authors studied the role of c-Met+ human CTLs in a three-dimensional setting. They made the observation that c-Met+ human CTLs showed an increased cytolytic activity, similar with their murine counterparts. The work here was just a simply repeat of their previous work, only in human CTL clone. Below are some points that should be addressed to improve its quality.
1. They should ask a native English speaker to edit it, since it contains several grammar mistakes. For example, “we demonstrated the immunosuppressive role of HGF on c-Met+ CTLs both in vitro and in vivo”, it should be “we demonstrated the immunosuppressive role of HGF in c-Met+ CTLs both in vitro and in vivo”.
2. The authors did not specify how many times most of the experiments had been repeated.
3. In Figures 2 and 5, the authors used t-test to perform statistics analysis. However, Figure 2 B&C has more than 2 groups. Student’s t test was suitable for two-group analysis.
4. In Figure 4C, the authors did not conduct statistical analysis.
Comments on the Quality of English Language
Moderate editing of English language is required.
Author Response
- They should ask a native English speaker to edit it, since it contains several grammar mistakes. For example, “we demonstrated the immunosuppressive role of HGF on c-Met+ CTLs both in vitro and in vivo”, it should be “we demonstrated the immunosuppressive role of HGF in c-Met+ CTLs both in vitro and in vivo”.
Answer: We would like to thank reviewer 1 for reviewing our article and providing constructive comments. As suggested, we had the text edited by a native English speaker
- The authors did not specify how many times most of the experiments had been repeated.
Answer: We have added what was missing, you'll find the changes in the figure legends in a character highlighted in yellow.
- In Figures 2 and 5, the authors used t-test to perform statistics analysis. However, Figure 2 B&C has more than 2 groups. Student’s t test was suitable for two-group analysis.
Answer: thank you for your comment. We have redone the statistical analysis using one-way ANOVA with Tukey's multiple comparisons test. We have made these corrections to figures (2B and C) and (5B and C) as well as to the figure legends.
- In Figure 4C, the authors did not conduct statistical analysis.
Answer: we apologize for forgetting to add the following detail; each point corresponds to the value of the mean of the triplicates. We have redone the statistical analysis using one-way ANOVA with Tukey's multiple comparisons test. We have made these corrections to figure 4C as well as to the figure legends.
Reviewer 2 Report
Comments and Suggestions for Authors
The manuscript presented by the authors contained important information about the role of c-MET and HGF on tumor immunity and examined their efficacy in vitro. Methods are sound and well presented, as well as results.
The main issues needed to be addressed by the authors are the structure and the extent of the discussion. The study described important results, but there are not discussed in a thorough way. Especially, the clinical correlation of c-MET+ CTLs with better antitumor effects, which is contrast with MET inhibitors, is confusing for the readed. Also, the result about decreased PD-1 or CTL-4 expression in c-MET+ CTLs should be further discussed.
Limitation of the study and they might be added by the authors after review.
Overall, the study is well-conducted, but with the need of further discussing the results to support its clinical value.
Comments on the Quality of English Language
minor editing needed
Author Response
Answer: we thank reviewer 2 for highlighting the quality of this work. We have improved the discussion in relation to the results obtained, and clarified their importance and relevance in clinical anti-tumor responses. We reworded the entire discussion section accordingly.
Reviewer 3 Report
Comments and Suggestions for Authors
The results of the work presented in this manuscript summarize novel, interesting and scientifically valuable informations. It seem sto be a timely and useful contribution.
The manuscript is well written and provides a lot of interesting data supported by various experiments and methods
After some minor corrections I suggest to accept the manuscript for publication.
Comments:
Thes should provide more detail about the cells and cell culture focusing on transgenic mice model.
The Discussion section should be extended giving more highlights.
Comments on the Quality of English Language
Seems to be fine.
Author Response
They should provide more detail about the cells and cell culture focusing on transgenic mice model.
Answer: we thank reviewer 3 for his pertinent remarks and overall positive comments. Details of spleen cell culture in Pmel mice are provided in the "CTL stimulation" paragraph of the method section, right after the paragraph on cells and culture.
The Discussion section should be extended giving more highlights.
Answer: as suggested, we have added more details in the discussion section and reworded the entire discussion section accordingly.
Round 2
Reviewer 1 Report
Comments and Suggestions for Authors
accepted
Author Response
We would like to thank reviewer 1 for agreeing to review our paper and for all the pertinent comments which helped to improve the quality of our work